# Vision-Based Robotic Grasping of Reels for Automatic Packaging Machines

**Simone Comari** and **Marco Carricato** *

Department of Industrial Engineering, University of Bologna, Viale del Risorgimento 2, 40136 Bologna, Italy
* Correspondence: marco.carricato@unibo.it; Tel.: +39-051-2093443

**Featured Application: The presented work is tailored to industrial applications in the field of automatic machines, in which handling of raw material reels is a common task, often tackled by human operators by hand. The employment of a robotic system equipped with a simple vision sensor (e.g., 2D camera) and a low computational power external unit is a viable solution to relieve operators from this strenuous job, while keeping the productivity rate high. In general, the procedure presented in this paper for the robust detection of reel cores by a vision system can be useful in any industrial process in which reels are involved and hardware resources are limited.**

**Abstract:** In this work, we present a vision system particularly suited to the automatic recognition of reels in the field of automatic packaging machines. The output of the vision system is used to guide the autonomous grasping of the reels by a robot for a subsequent manipulation task. The proposed solution is built around three different methods to solve the ellipse-detection problem in an image. Such methods leverage standard image processing and mathematical algorithms, which are tailored to the targeted application. An experimental campaign demonstrates the efficacy of the proposed approach, even in the presence of low computational power and limited hardware resources, as in the use-case at hand.

**Keywords:** computer vision; ellipse detection; robotic grasping; automatic machine





## 1. Introduction

Two-dimensional vision-based algorithms for object detection and pose estimation have been widely studied and employed in industry for decades [1], mainly thanks to the fact that they solely rely on images, a source of information easy to obtain with relatively low costs. Although prices may depend on the quality of the adopted camera, powerful open-source software libraries such as *OpenCV* [2] or *scikit-image* [3] are publicly available and easy to use, and integrate into most computer-driven applications. Leveraging the most common algorithms for image processing and computer vision (CV), many companies developed visual applications, tools, and smart devices [4] that allow end users to often utilize this powerful technology without any expertise on the subject.

Although standard CV approaches have been and still are heavily employed in production lines of the manufacturing industry [5,6], such as in quality control [7] or part identification [8], they become less trustworthy in unstructured environments, where targets may be substantially different in shape, texture and size, and exist in highly variable conditions. A breakthrough in this field occurred with the introduction of machine learning (ML), and, further on, deep learning (DL) for image analysis and elaboration [9–11]. Thanks to their inherent versatility, coming from the fact that they are purely data driven, robust solutions were developed that well suit even those highly variant scenarios where standard CV was prone to failures. In [12], the authors provide a comprehensive review of significant CV techniques applicable to the manufacturing field, with a special focus on the most recent DL-based approaches. In their critical analysis, they also outline some long-standing issues

and challenges related to the complexity of implementation and integration of most recent findings in long-established companies, the problem of data collection, preprocessing and labelling, and the difficulty of defining benchmarks that can be applied to different scenarios.

As a consequence, care must be taken before switching to ML/DL approaches, because they typically require powerful computing units, may be too slow for some production lines, do not yet easily integrate with other industrial devices such as PLC, and need plenty of labelled and uniform data samples to be efficiently trained to work correctly and robustly.

It is the integrator's role to select the best option given the context at hand. In particular, for structured and repetitive environments, it is often a better choice to rely on standard approaches that still provide fast and trustworthy results with limited resources.

The use-case at hand lies in this context.

### 1.1. Related Works

When it comes to part manipulation using a robotic arm, CV may play a major role in increasing the flexibility and productivity of the industrial process in which the robot is employed. Robust and efficient grasping tasks (often called "bin-picking" tasks) have been studied for many years for this purpose.

Currently, many bin-picking systems utilize binocular vision methods that combine feature point matching and projection geometry to estimate the location of an item. Successful work in this context is presented by Oh et al. [13] who propose a geometric pattern matching method to find the correspondence points in stereo images, and improve the selection of pick-up candidates. Standard geometric features such as circles and squares are common in the manufacturing field. For this reason, a lot of research has been dedicated to solving these particular problems. An example related to circular holes can be found in [14], wherein the authors improve the work by Malassiotis and Strintzis [15] by providing a more robust method to find edges in the presence of interference from the inner wall of a hole, and by introducing epipolar constraints to fit ellipses in the two views simultaneously.

Three-dimensional sensors, however, still have some limitations, such as the higher cost with respect to traditional 2D cameras, their high sensitivity to calibration, and the narrow field of view shared by the stereo cameras, which makes it difficult to have the same feature points always captured by both cameras, especially during the grasping process, in which there can be an occlusion due to the picked object. In [16], the authors propose an alternative by using a 2D monocular vision for part localization and grasping, leveraging a highly engineered calibration and matching algorithm. In [17], a single-shot multi-box detector is used to quickly define the region of interest in which to look for contour features and estimate the pose of the known object by perspective-n-point (this pertains to the problem of estimating the pose of a calibrated camera given a set of $n$ 3D points in the real world and their corresponding 2D projections in an image). With reference to circular object localization, to which this article mainly relates, we mention the work from Luo et al. [18], in which the authors present a continuous edge detector inspired by the Canny detector, and a fast ellipse detector by using the randomized Hough transform (RHT). A work based on more recent DL techniques presented by Liu et al. [19], finally, claims to outperform state-of-the-art approaches in ellipse detection in terms of precision (the ratio between the number of true positive results and the number of all positive results (including false positive results), recall (the ratio between the number of true positive results and the number of all samples that should have been identified as positive), and F-measure (a measure of a test's accuracy calculated from the precision and the recall of the test) on industrial images.

Despite the many successful examples in the literature, there is still a gap between the implementation in a controlled environment, such as a research laboratory, and the implementation in an actual industrial scenario. Manufacturing companies are often reluctant to invest in generic vision systems that claim to adapt to a large number of conditions and prefer to adopt a more specific solution that can guarantee a success rate

close to 100% even if it is less flexible or elegant. By following this reasoning, this paper aims at bridging the gap by bringing together a set of solutions that, when combined, attain the desired level of industrial reliability. Rather than presenting a novel method to detect an ellipse through monocular vision, our goal is to show how a suitable combination of existing techniques can be put together to achieve impressive results in a real-case scenario.

### 1.2. Use Case: Reel-Picking for Automatic Packaging Machines

In automatic packaging machines, raw material often comes in the form of reels, wrapped around a cardboard core of standardized dimensions (e.g., diameter and thickness). Although the sizes of the core are known, in a real production context the overall diameter of the reel and the raw material texture and color may differ from one batch to the other.

In the use-case analyzed for this work, three types of reels are employed, whose main features are reported in Table 1. The reels are stocked in four semi-horizontal piles on a mobile deposit, called "wagon", as shown in Figure 1. At the beginning of each shift, the operator brings the wagon to the warehouse, loads it with as many reels as it can store, and moves it back to its original position on the shop floor. A mobile robot is then in charge of picking a single reel and loading it onto the automatic machine that requested it. The mobile robot is equipped with a robotic arm, an electrical gripper with custom fingers for inside grasping, and an eye-on-hand industrial camera [20]. An industrial PC is also installed onboard the mobile robotic platform and takes care of reading the camera stream, elaborating the incoming images, and communicating with the robot through REST APIs (web Application Programming Interfaces -APIs- or services conforming to the REpresentational State Transfer -REST- architectural style).

**Table 1.** Reels features.

| Reel Type | ID | Core Diameter (Inner/Outer) | Thickness |
|---|---|---|---|
| *Paper* | 1 | 77/85 mm | 100 mm |
| *Tags* | 2 | 158/166 mm | 30 mm |
| *Outer Envelope* | 3 | 77/85 mm | 100 mm |

The objective of this work is the development of a robust vision-based algorithm that, employing standard CV techniques and math libraries, can infer from a single image the 3D pose of a reel core with respect to the camera frame. With this information, and knowing from calibration the position of the camera with respect to the end-effector of the manipulator, it is then possible to express such pose in the robot base frame. Finally, a suitable target configuration for the robotic arm may be calculated to insert the gripper fingers into the core, and perform the grasping of the reel (from inside the core).

The specifications given for this industrial task are:

1. finger insertion in reels with ID 1 and 3 (see Table 1) require millimetric accuracy due to the limited gap between the fingers and the inner diameter of the core (e.g., 9 mm);
2. it must work with gray-scale 1544 × 2064 p images;
3. it must run on an industrial workstation running Ubuntu Server 18.04 LTS operating system and limited computing power (e.g., Intel(R) Core(TM) i5-6440EQ CPU @ 2.70 GHz, 8 GB RAM, no GPU, no graphics card); and
4. it must execute within 3.0 s.

The proposed solution relies on a reel-based selection policy of three different methods to extract the contour of the reel core framed from a single image. Next, by exploiting the knowledge of the core's inner/outer diameter dimension, the algorithm infers its 3D pose with respect to the camera.

The article is structured as follows. Section 2 describes how to obtain the 3D pose estimate of a reel from a digital ellipse that represents the contour of its core in a picture. In Section 3, three methods to obtain such an ellipse from a single image are proposed, all of

them based on known techniques suitably put together. In Section 4, an effective method-selection policy is described that optimizes the results that are presented in Section 5, obtained from the data collected in an experimental campaign. Lastly, in Section 6, conclusions are drawn.

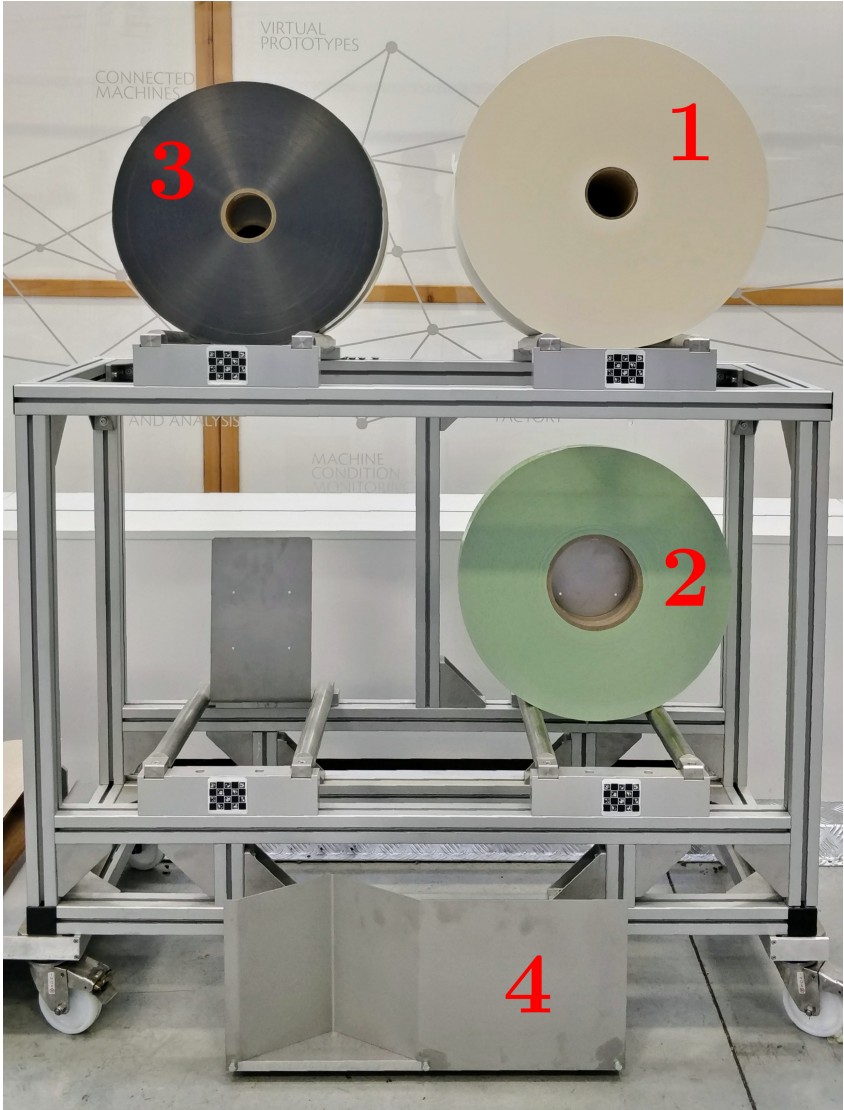

**Figure 1.** Wagon and reels. (1) Filter-paper reels on the top-right shelf, (2) tag reels on the bottom-right shelf, (3) outer-envelope reels on the top-left shelf, (4) VL-marker at the bottom for AGV docking.

## 2. Reel-Pose Estimation

The three-dimensional circle-pose estimation from a 2D ellipse has been widely studied for its practical usefulness in several fields such as eye-direction recognition and localization of round-shaped objects. In [21], an analytical approach is given exploiting the geometrical properties of a projection camera. This idea relies on the evidence that according to the chosen perspective (i.e., reference camera frame), an oblique circular cone and an oblique elliptical cone provide the same projection of a circle onto an ellipse in 3D space and represent the same surface. Starting from a 2D ellipse detected on a single image, and given both the camera intrinsic parameters and the corresponding circumference radius in the real world, the outcome of this algorithm consists of a 3D point representing the center of the circle expressed in the camera frame together with a versor normal to the plane the circle belongs to. However, it is important to notice that, from a mathematical point of view, two valid solutions are always obtained as an outcome of this formulation. Intuitively,

due to its symmetrical properties, it is impossible to distinguish between the two possible "directions" that can provide the same projection. A schematic view of this ambiguity is shown in Figure 2.

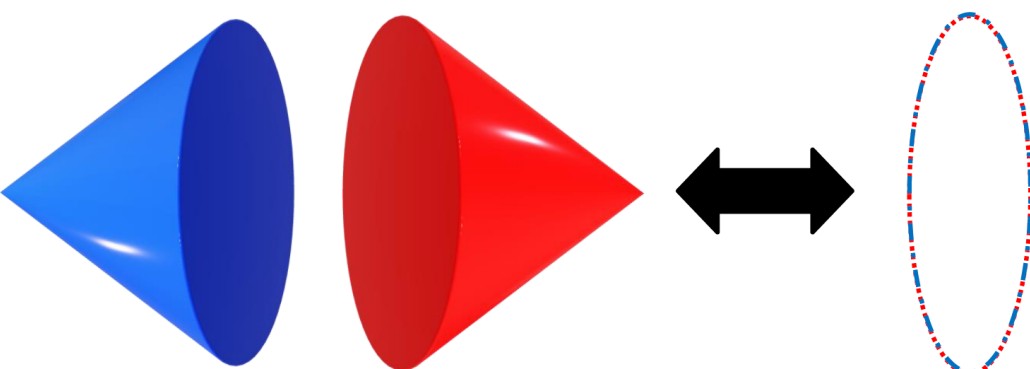

**Figure 2.** Circle projection ambiguity.

To resolve this issue, we exploit the knowledge of the relative side-positioning of the camera with respect to the framed circle, that is either right or left, which allows us to select the only realistic solution between the two.

The main assumption of the algorithm developed for ellipse detection is that a single reel core is always present within the image taken by the camera. This holds thanks to the fact that the mobile robot can approach the reel's deposit with a certain accuracy by detecting a 3D VL-shaped marker installed at its bottom (see Figure 1), which is easily recognizable by the laser scanners aboard the platform. Once this approximate relative positioning is defined, a suitable framing configuration (relative to the wagon) can be manually determined once for all for each pile to guarantee that the core of each reel in the stack is visible inside the image. In other words, the cone of view of the camera should include all the possible positions of the reel cores for the given pile. Alternatively, if the field of view of the camera is too narrow to frame all the cores from a single perspective, one could foresee a scanning motion that covers the whole length of the pile. In any case, the camera perspective should be chosen such that the circular section of the reel core appears as a slender ellipse, in order to reduce the noise-to-signal ratio when back-projecting from 2D to 3D coordinates.

## 3. Reel Core Detection

After extensive research about available algorithms and functions for ellipse detection and estimation, three methods were selected, adjusted, and rearranged in a pipeline that aims at covering most of the cases that can practically occur. The reason behind the choice of having multiple approaches instead of a single one is dictated by a trade-off between robustness and speed: although some methods may work well and fast for some specific scenarios, others might be in general more effective and robust, but take an unnecessarily long time. In a nutshell, we address the selected methods as follows:

1.  *Contour method*: relies on closed contours, which ideally should coincide with the target ellipse; it is very fast, and it works well in the presence of high contrast between the core and the surrounding wrapped material;
2.  *RBrown method*: relies on edges including portions of the target ellipse; it is discreetly fast, and it works even in low-contrast conditions, but it is prone to false positives (in our context, a false positive is the detection of a wrong ellipse, which does not coincide with the true reel-core contour);
3.  *Randomized Hough transform*: relies on edges including the extremes of the target ellipse along the major axis direction; it is much slower than previous methods, but it is more robust even in low-contrast conditions.

### 3.1. Ellipse Detection by Contour Method

This method relies on the evidence that when the core is much darker than the surrounding material, the closed contour which encapsulates it is itself the target ellipse, corresponding to the external border of the cylindrical reel core. To extract the contour, the image needs to be converted to a binary form, after the application of a threshold, which maps pixels darker than a certain value to black and the remaining ones to white. Due to different light conditions, this threshold is dynamically tuned by inspecting the histogram of the grayscale original image. Moreover, to get rid of noisy features like dots and black spots, the original image goes through several blurring phases until a proper contour is found. Nevertheless, due to imperfections on the real border, this contour can present small flaws or unexpected edges, which, however, are usually non-significant. Hence, a proper contour is defined according to the area of the surface it encapsulates, and to the "goodness" of the ellipse fit through the least-square method, given by the residual error between the true contour points and the points of the estimate. Figure 3 depicts the data flow for the described method, whereas Figure 4 illustrates the main steps.

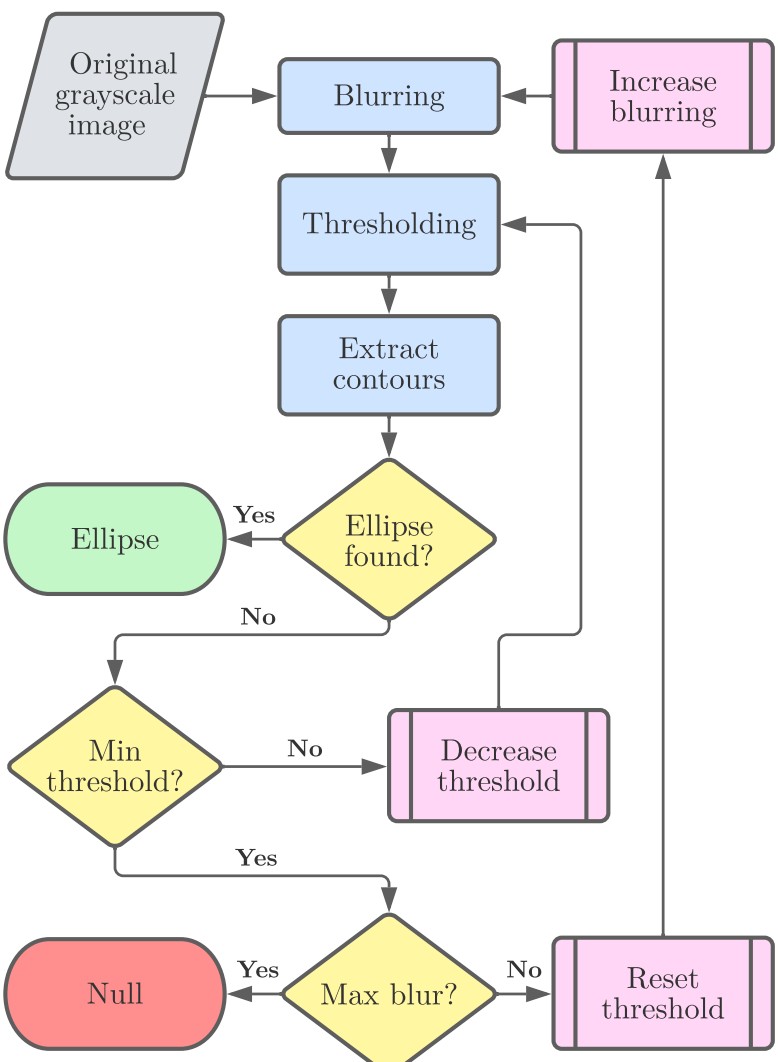

**Figure 3.** Contour method flowchart.

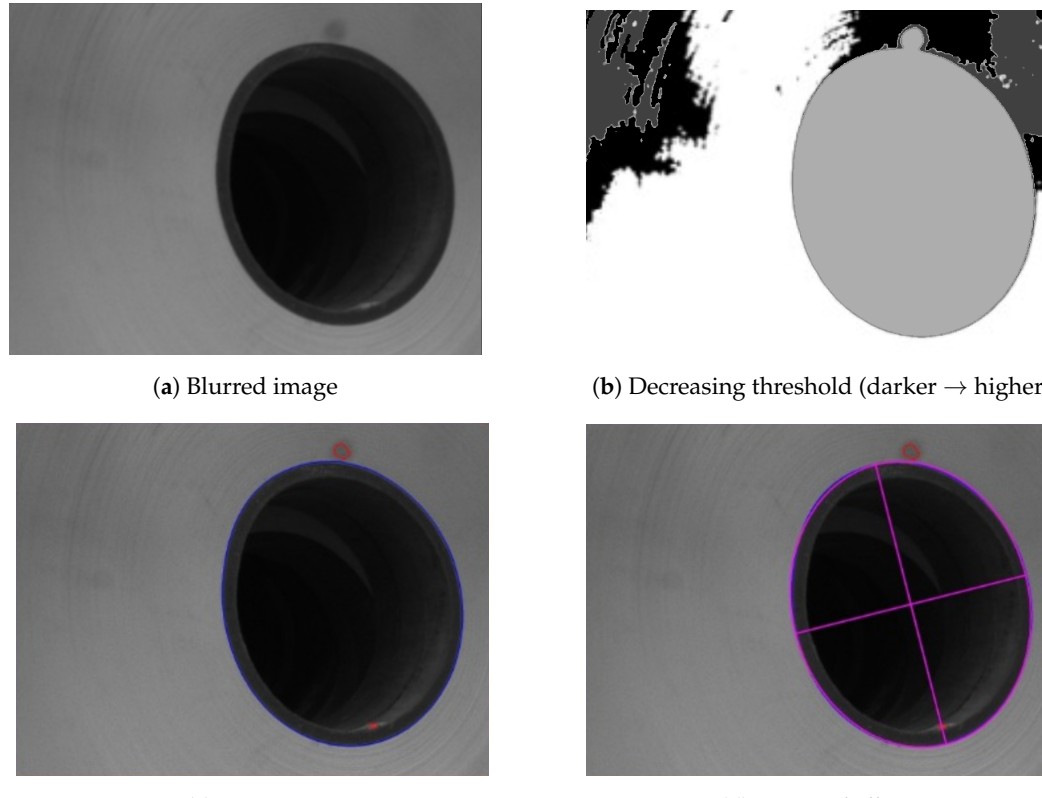

**Figure 4.** Contour method steps.

### 3.2. Ellipse Detection by RBrown Method

It is not always possible to extract a clean, closed contour out of an image. Due to adverse light conditions, the contrast between the core and the surrounding material might be very low and lead to an impossible distinction between the two parts, sometimes even for the human eye. Employing edges instead of contours, we switch the focus of the algorithm to actively close a whole ellipse starting from a single solid portion of it. This method is also effective for those cases in which the ellipse is not completely visible due to occlusions or for being partially out of frame, rather than for low-contrast reasons. Nonetheless, it is prone to produce false positives whenever the portion under consideration has a small curvature (i.e., close to the extremes of the ellipse's minor axis). On the contrary, it provides very good results when the arc includes the location of maximum curvature points, which correspond to those points close to the extremes of the major axis. This is related to the noise-to-signal ratio, which amplifies errors given by noisy edges when the curvature is low and vice versa.

From a process point of view, the first step is the extraction of proper edges. Parameters of standard functions for edge detection are dynamically tuned so as to produce a minimum number of solid edges of a given minimum length (in pixels). Moreover, close-by edges are actively connected by binary closing operations (*closing* is a mathematical morphology operation that consists in the succession of dilation and an erosion of the input with the same structuring element, therefore, filling holes smaller than the structuring element) and skeletonization (the *topological skeleton* of a shape is a thin version of that shape that is equidistant to its boundaries).

For each edge in the set, two ellipse-fitting functions, respectively derived from the work of Brown [22] and Gander et al. [23], are invoked and, therefore, up to two candidate ellipses are generated:

1.   a linear least-square-ellipse fit using the Bookstein constraint; and
2.   a non-linear (Gauss-Newton) least-square-ellipse fit based on an initial guess given by linear least-square-ellipse fit using trace constraint.

Both candidates for every single edge are validated through the same "goodness" check presented in Section 3.1.

The collection of candidates of all edges found is then optionally filtered by using additional information provided by the user, such as the expected ranges for aspect ratio or major-axis length. A simple outlier removal based on centroids and aspect ratios is used to refine the collection. Finally, the smaller or larger ellipse among all candidates is taken according to the reel type (from experience, with reference to Table 1, for ID = 1 or 2, the larger ellipse coincides with the external border of the reel core, whereas for ID = 3 the smaller ellipse coincides with the more evident internal border). A schematic view of the mentioned process is displayed in Figure 5, whereas Figure 6 shows the main steps.

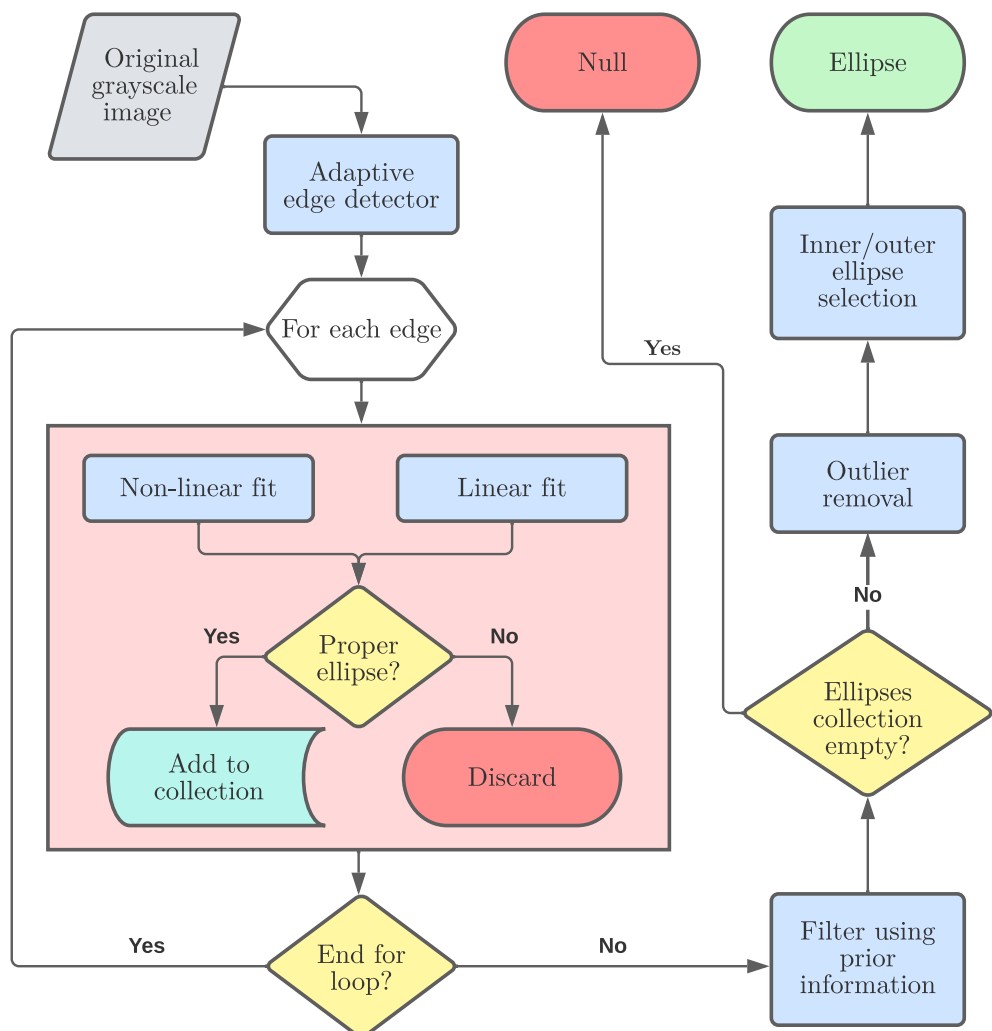

**Figure 5.** RBrown method flowchart.

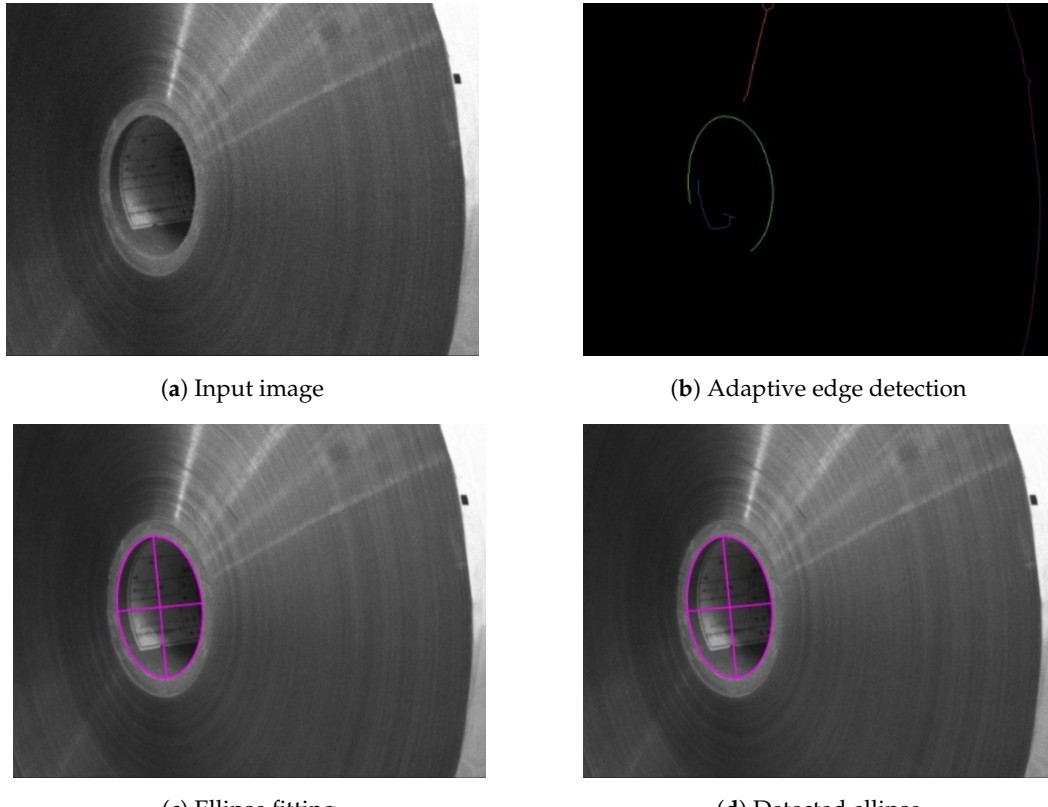

(**a**) Input image

(**b**) Adaptive edge detection

(**c**) Ellipse fitting

(**d**) Detected ellipse

**Figure 6.** RBrown method steps.

### 3.3. Ellipse Detection by Randomized Hough Transform

Like the RBrown method presented in Section 3.2, the Hough transform also relies on edges instead of contours but, on the contrary, it uses all of them at once to find the best fit.

The algorithm is a direct implementation of [24,25], and it assumes that both extremes of the major axis are present within the edges. Being a loop-based algorithmic implementation, the key to improving the speed of this computationally expensive process are filters and randomization (i.e., random sub-sampling of original data).

Once again, prior knowledge of the target ellipse drastically reduces the number of iterations and, therefore, is strongly recommended to exploit information such as the expected dimension and shape (e.g., AR).

Briefly, each iteration picks a pair of two points among the edges and tries them as extremes of the major axis. Moreover, because we work with edges and not sparse points, each point belongs to a curve and therefore can be assigned a tangent-angle value, in the range ($-90°$:$90°$]. Exploiting these two facts, we can further restrict the selection of pairs to those points that share similar tangent angles, being de facto two opposite points in the ellipse.

A scheme of the data flow for this last method is shown in Figure 7, whereas Figure 8 illustrates its main steps.

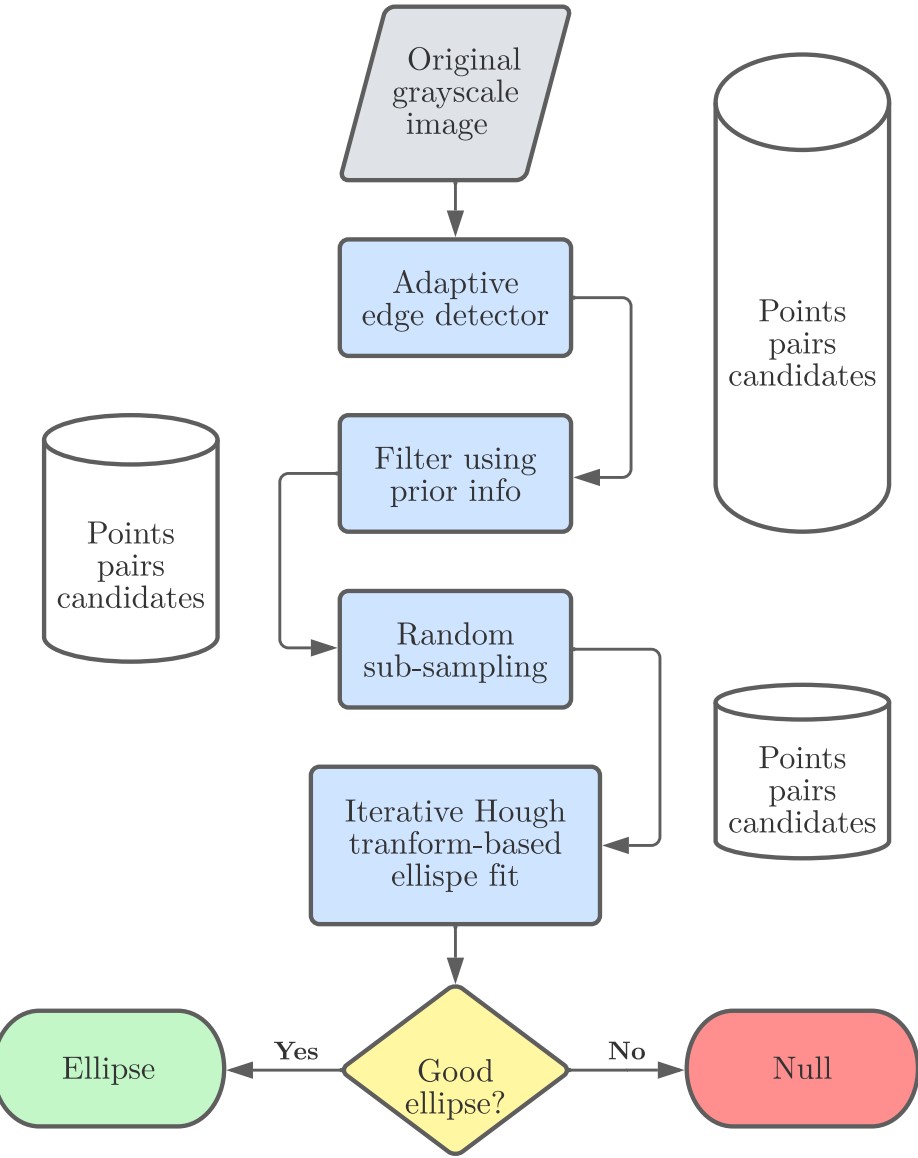

**Figure 7.** Hough transform method flowchart.

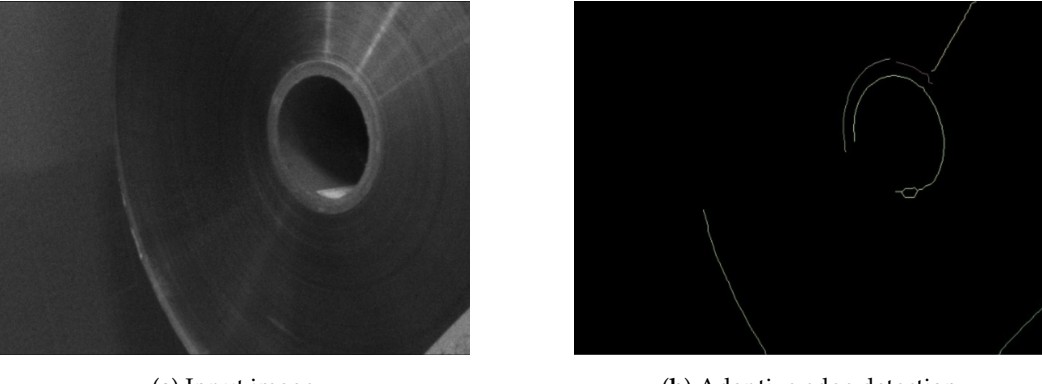

(**a**) Input image        (**b**) Adaptive edge detection

**Figure 8.** *Cont.*

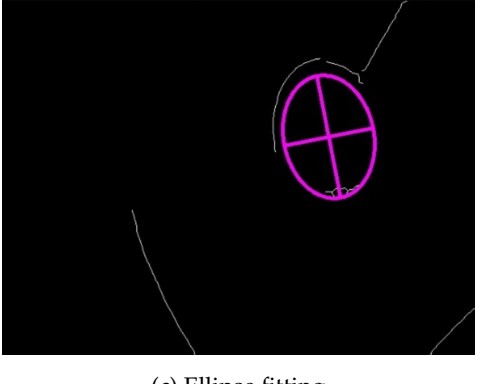
(**c**) Ellipse fitting

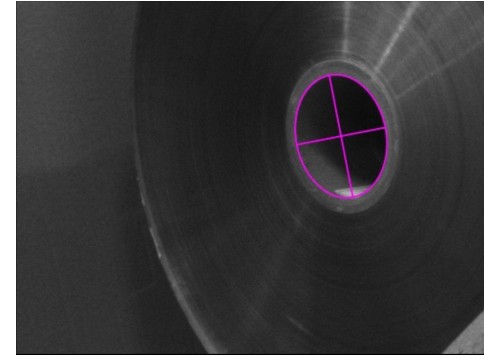
(**d**) Detected ellipse

**Figure 8.** Hough-transform method steps.

## 4. Method Selection Policy

From previous consideration and experience, we outlined a selection flow for picking the best method according to the different characteristics of each reel. In particular, paper and tag reels (ID = 1, 2) show high contrast between the core and the surrounding wrapped material and, therefore, are suitable for the contour method, which is also the fastest of the three. RBrown and Hough transform are thus kept as successive fallback options in case the first method fails, being more robust, but also slower. The outer envelope reel is highly reflective, and it is impossible to clearly extract a closed contour from a grayscale image. For this reel type, the best performing method is the Hough transform, and RBrown is kept as a fallback option in those rare cases when it fails (for instance when both major-axis extremes do not appear in the edges). Additionally, to speed up computation time, for Rbrown and Hough transform methods a downsized version of the original image is used (resized to 480p) and several tests showed a negligible degradation of performance in terms of accuracy, sometimes even leading to improvements. Moreover, because Rbrown and Hough transform methods share the same starting phase, that is edge detection, in case of failure, extracted edges are reused by the next method to save uptime. A schematic view of the selection policy is illustrated in Figure 9.

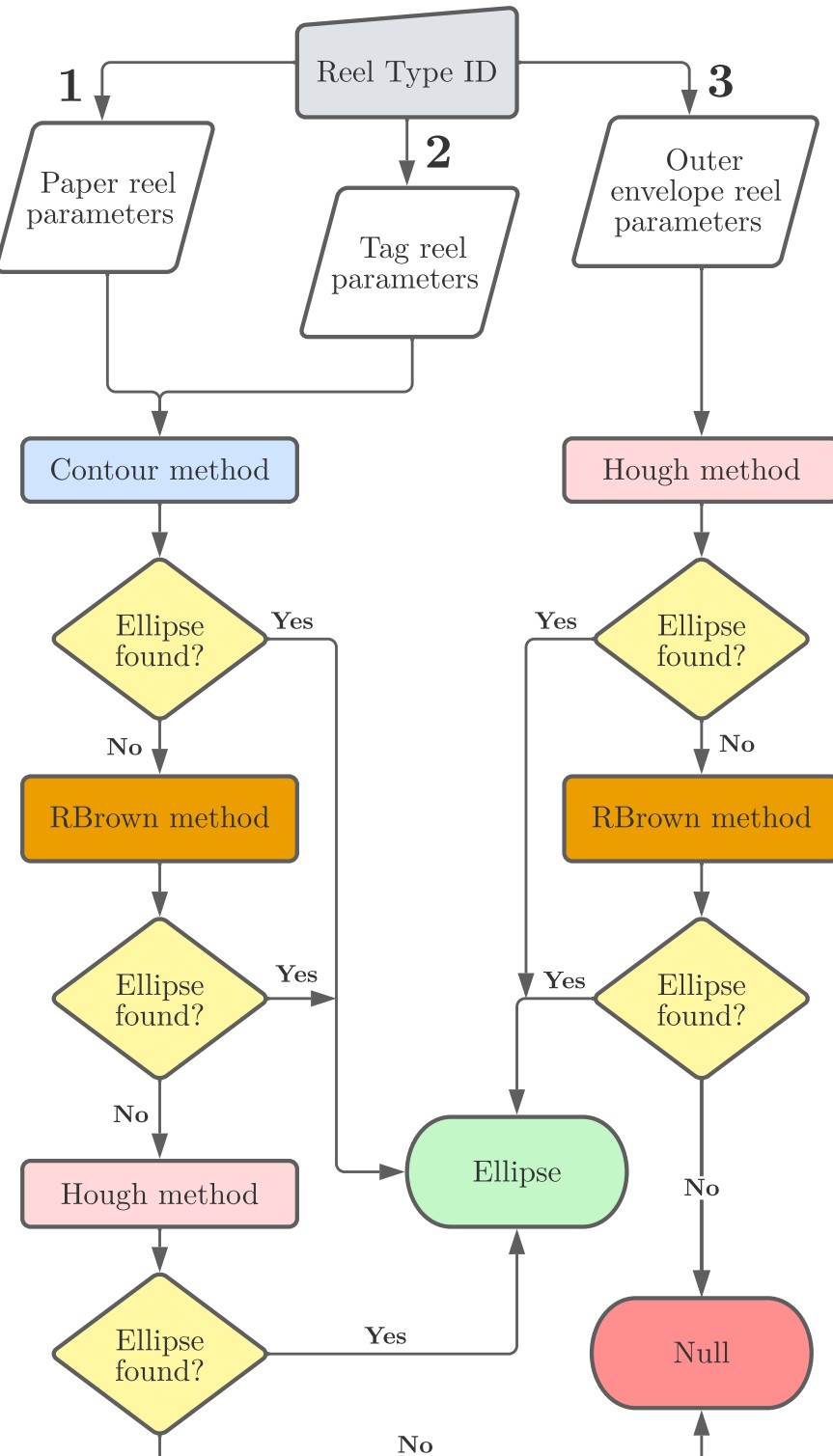

**Figure 9.** Method selection flowchart.

## 5. Experiments and Results

To test the efficacy of the proposed methods, we assigned the mobile robot the task of picking each reel many consecutive times. Specifically, the operation consists in approaching the wagon, scanning the targeted pile from a predefined perspective (discussed in Section 2) and, if found, inserting the gripper in its core for an inside grasping, as shown in Figure 10.

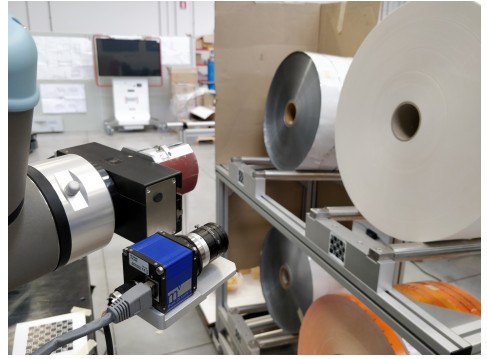

(**a**) Framing

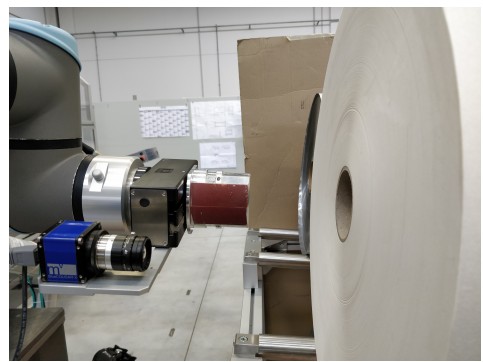

(**b**) Approach

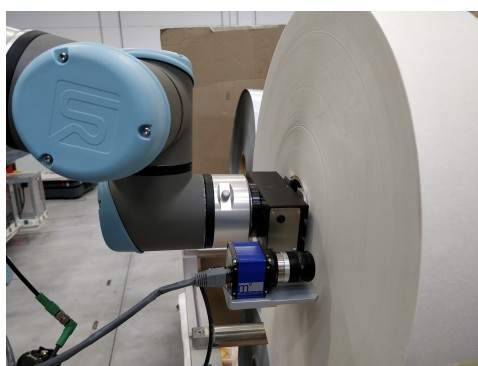

(**c**) Insertion

**Figure 10.** Picking preliminary operations.

The pose estimation is considered successful if no issues are detected during the insertion phase. In fact, once the reel pose is known, the robot moves its tool-center point (TCP) (in this case, located at the tip of the gripper fingers) 10 cm in front of its center, and, to insert the gripper fingers, moves linearly toward the reel in the *Z*-direction of its reference frame (i.e., the longitudinal axis of the cylinder that represents the reel core) until contact. Because the thickness of the reel is known, given the predefined approach distance and the geometry of the gripper, it is possible to guess the expected length of the path up to flushing. If the estimation is not exact, due to the little clearance between the size of the gripper fingers and the diameter of the reel, the gripper will hit the reel before running the whole insertion distance, resulting in a failure. On the contrary, if a contact is not detected beyond the expected insertion length, an error is raised.

The wagon approach operation introduces a small uncertainty due to the positioning error of the mobile platform with respect to the stationary wagon, typically less than 1 cm in position and 2° in orientation. This inconvenience is exploited here to enhance the robustness of the process against small changes in the resulting framing perspective, which is computed in the mobile robot frame. Moreover, reels at different locations on the shelf, from the one at the front to the one at the back, were used during the test to avoid biases derived from their position in the pile.

Table 2 shows the results of the whole experimental campaign in terms of computational time and final outcome.

**Table 2.** Experiments results.

| Reel Type | Total Trials | Average Time | Maximum Time | Failures |
| --- | --- | --- | --- | --- |
| *Paper* | 109 | 0.749 s | 0.857 s | 0 |
| *Tags* | 118 | 0.786 s | 1.445 s | 1 |
| *Outer Envelope* | 55 | 3.831 s | 4.392 s | 0 |

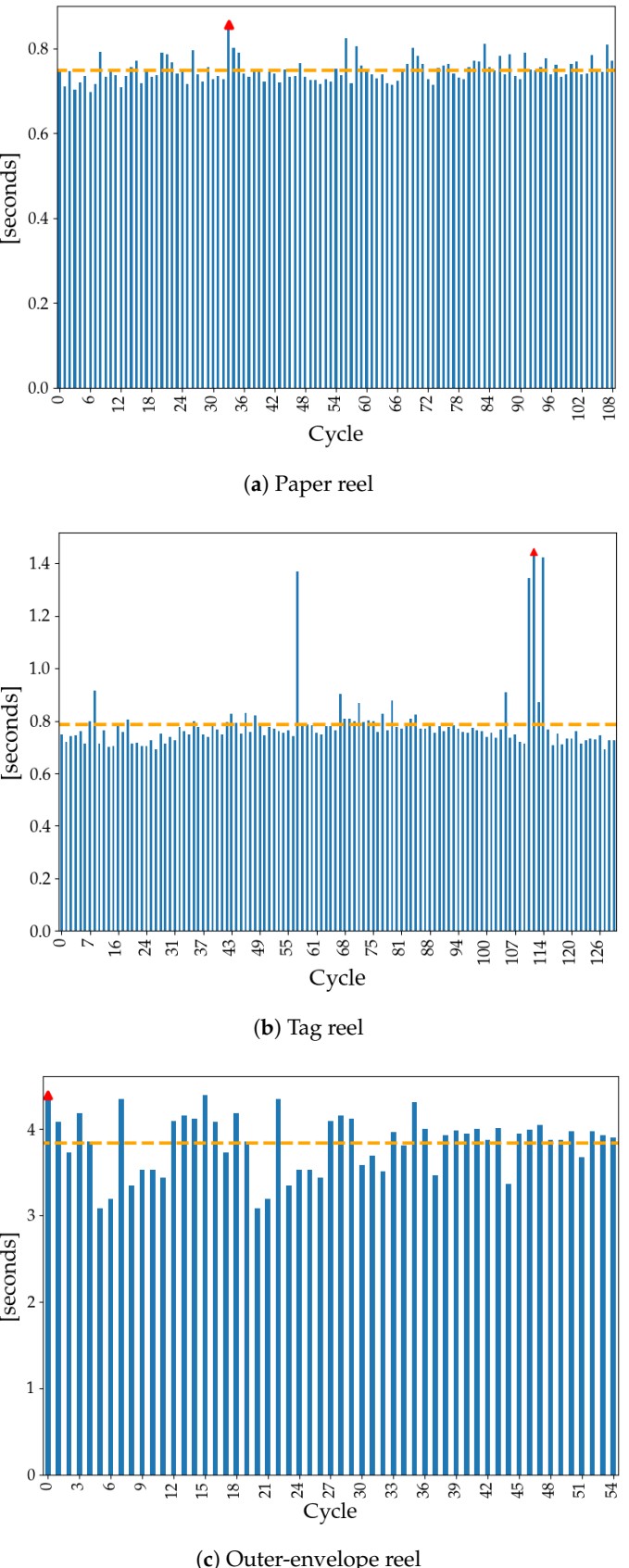

(**a**) Paper reel

(**b**) Tag reel

(**c**) Outer-envelope reel

**Figure 11.** Computational times for reel-core-pose estimation. The dashed line represents the average time and the red marker the maximum time experienced (see Table 2).

The contour method was successfully employed in all cycles of paper-reel and tag-reel detection, except for a single iteration in the latter one, in which none of the proposed approaches led to success. For the outer-envelope reel, instead, the Hough method proved to be the most suitable given the zero failures achieved. As expected, the contour method is the fastest, allowing the system to provide a pose estimate in less than 1 s on average, as shown in Figure 11. The largest values are related to an adverse light condition that pushed the algorithm to autotune the parameters before computing a valid response, as described in Section 3.1. The Hough method, despite being very effective and robust, is also the slowest by a large amount. The iterative procedure that looks over all possible combinations in the given (reduced) search space introduces a computational burden which is hard to avoid. As a consequence, the time constraints for this particular reel could not be satisfied (the target was 3.0 s) even if by a relatively small amount on average. However, the outer-envelope reel is the one which takes longer to be consumed by the automatic machine. Therefore, a little extra time is still acceptable as it does not significantly impact the overall reel-changing time.

*Tuning and User-Defined Settings*

The excellent outcomes achieved were the result of a fine-tuning session that allowed us to be particularly robust in the current scenario. Despite the autotuning of a few parameters to cope with varying light condition, there is a set of fixed parameters which must be suitably defined by the user. They do not require any special skill or expertise and, thus, can be easily managed by a non-expert operator. However, they do need a short trial-and-error session as they are strictly related to the expected kinds and variability of framed reels. These settings are mainly used to discard potential false positives, and restrict the search space over valid outcomes, but also to speed up computational time (especially the ones used by the Rbrown and Hough method). In accordance with this concept, they express ranges of valid values for the detected ellipse in the image rather than single reference values. In Tables 3–5, the settings used for this experimental campaign are reported.

**Table 3.** Experiments settings for filter-paper reel.

| Name | Value | Used by (C: contour, R: RBrown, H: Hough) | Description |
|------|-------|-------------------------------------------|-------------|
| *min_major_axis* | 400 | R,H | Minimum major-axis length in pixels of the ellipse resembling the reel core. |
| *max_major_axis* | 1100 | R,H | Maximum major-axis length in pixels of the ellipse resembling the reel core. |
| *min_aspect_ratio* | 0.7 | R,H | Minimum aspect ratio of the ellipse resembling the reel core. |
| *max_aspect_ratio* | 1.0 | R,H | Maximum aspect ratio of the ellipse resembling the reel core. |

**Table 4.** Experiments settings for tag reel.

| Name | Value | Used by | Description |
|------|-------|---------|-------------|
| *min_contour_area* | $3 \times 10^5$ | C | Minimum area in pixels$^2$ circumscribed by the contour of the ellipse resembling the reel core. |
| *max_contour_area* | $3 \times 10^6$ | C | Maximum area in pixels$^2$ circumscribed by the contour of the ellipse resembling the reel core. |
| *max_goodness* | 15 | C | Maximum value of "goodness" for a valid ellipse. |
| *min_major_axis* | 1000 | R,H | See Table 3. |
| *max_major_axis* | 1400 | R,H | See Table 3. |
| *min_aspect_ratio* | 0.7 | R,H | See Table 3. |
| *max_aspect_ratio* | 1.0 | R,H | See Table 3. |
| *rotation* | 70 | R,H | Rotation of the ellipse wrt horizontal axis in degrees. |
| *rotation_span* | 15 | R,H | Tolerance for *rotation* in degrees. |

**Table 5.** Experiments settings for outer-envelope reel.

| Name | Value | Used by | Description |
|------|-------|---------|-------------|
| *min_major_axis* | 1000 | H,R | See Table 3. |
| *max_major_axis* | 1400 | H,R | See Table 3. |
| *min_aspect_ratio* | 0.7 | H,R | See Table 3. |
| *max_aspect_ratio* | 1.0 | H,R | See Table 3. |
| *rotation* | 70 | H,R | See Table 4. |
| *rotation_span* | 15 | H,R | See Table 4. |
| *min_edge_size* | 100 | H,R | Minimum candidate edge length in pixels. |
| *sigma0* | 4.0 | H,R | Initial blurring value for smoothing the image before edge detection. |
| *min_feat* | 4 | H,R | Minimum number of detected edges. |

## 6. Conclusions

In this article, we presented a method to robustly estimate the 3D pose of a raw-material reel from a single image and prior information for picking purposes in an industrial environment.

The work was justified by the industrial need of implementing a vision-based system for allowing a mobile robot equipped with a manipulator to autonomously pick three different types of reels from a deposit, and load them onto an automatic packaging machine. The envisioned solution was implemented because it was easy to share, document, maintain, and it was deployable on an industrial PC with limited power resources.

Three methods for detecting a single ellipse out of a picture framing the reel core were proposed (i.e., contour, RBrown, and RHT), and the results coming from an extensive experimental campaign showed how, following an optimal selection policy, the success rate touched 99.6%. We showed how the different photometric characteristics of each reel affect the efficacy of each method. In particular, the most selected method was the contour one, used for the detection of both paper and tag reel, which presented the highest contrast between the reel core and the material wrapped around it. For the outer-envelope reel, which has the lowest contrast and is highly reflective, the RHT method proved to be the most effective, but also the one with the slowest performance. The given approach is indeed influenced by a number of tuning parameters that can be manually changed according to the features of the reel under consideration.

The ellipse found in the image, which indicates the contour of the reel core, together with the prior knowledge of its real-world dimensions, is used to calculate the position of the reel center and the direction of its longitudinal axis, both expressed in the robot frame. This data is then sent to the robot to execute the picking of the object by inside grasping.

All specifications were met, with the only exception of time constraints for the case of the outer-envelope reel. Due to the intrinsic computational burden of the RHT calculation, in fact, the overall estimation process exceeds the target maximum execution time by 0.831 s on average. A possible improvement to speed up the process is porting the same software implementation from Python language to a compiled programming language, such as C++. Additionally, a better subsampling strategy and better filters may help reduce the number of iterations executed by the algorithm. Lastly, without limitations on the hardware and, thus, a better-performing computing unit, more sophisticated approaches based on the most recent DL techniques may be pursued and lead to similar or better results.

**Author Contributions:** Conceptualization, S.C.; methodology, S.C.; software, S.C.; validation, S.C. and M.C.; formal analysis, S.C.; investigation, S.C.; resources, S.C.; data curation, S.C.; writing—original draft preparation, S.C.; writing—review and editing, M.C.; visualization, S.C.; supervision, M.C.; project administration, M.C. All authors have read and agreed to the published version of the manuscript.

**Funding:** This research received no external funding.

**Institutional Review Board Statement:** Not applicable.

**Informed Consent Statement:** Not applicable.

**Data Availability Statement:** Not applicable.

**Acknowledgments:** The authors would like to thank the Research & Innovation dept. of IMA S.p.A. for providing the equipment and the use-case this work is built upon. In particular, we are grateful for their technical support, especially with regards to the communication with the industrial camera.

**Conflicts of Interest:** The authors declare no conflict of interest.

## Abbreviations

The following abbreviations are used in this manuscript:

| | |
|---|---|
| CV | Computer Vision |
| ML | Machine Learning |
| PLC | Programmable Logic Controller |
| RHT | Randomized Hough Transform |
| ID | IDentifier |
| PC | Personal Computer |
| LTS | Long-Term Support |
| CPU | Central Processing Unit |
| RAM | Random Access Memory |
| GPU | Graphic Processing Unit |
| API | Application Programming Interface |
| REST | REpresentational State Transfer |
| AR | Aspect Ratio |
| TCP | Tool-Center Point |

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
