# Peer review of "Vision-Based Robotic Grasping of Reels for Automatic Packaging Machines"

_applsci, doi:10.3390/app12157835_

Round 1
Reviewer 1 Report
In this article, we presented a method to robustly estimate the 3D pose of a raw- material reel from a single image and prior information for picking purposes in an industrial environment.
The work was justified by the industrial need of implementing a vision-based system for allowing a mobile robot equipped with a manipulator to autonomously pick three different types of reels from a deposit, and load them onto an automatic packaging machine.
The envisioned solution was implemented to be easy to share, document, maintain and be deployed on a industrial PC with limited power resources.
Three methods for detecting a single ellipse out of a picture framing the reel core were proposed (i.e. contour, RBrown and Hough transform), and the results coming from an extensive experimental campaign showed how, following an optimal selection policy, the success rate touched 99.6%. We showed how the different photo-metric characteristics of each reel affect the efficacy of each method.
The proposed solution relies on a reel-based selection policy of three different methods to extract the contour of the reel core framed from a single image. Next, by exploiting the knowledge of the core’s inner/outer diameter dimension, the algorithm infers its 3D pose with respect to the camera.
I found your article very interesting, but I suggest introducing following remarks, which have to be added and fulfilled before publishing the paper:
The Introduction section is very short. This section should contain The actuality of the problem, the current research revision in this subject area with the allocation of unsolved parts of the general problem, and finally, formulation of the research goal. Thus, this section should be rewritten.
The submitted article contains little -used literature (only 10 resources).
The conclusions section does not reflect the brief content of the manuscript, including the analysis and assessment of the results obtained.
In the submitted article, it is not listed in which the authors are planning to continue research.
After improving the above described issues in the paper I’d like to give my positive opinion on signing my review report.

Reviewer 2 Report
In this work, they present a vision system especially suitable for the automatic recognition of reels in the field of automatic packaging machines. The output of the vision system is used to guide the autonomous grasping of the coils by a robot for a subsequent handling task. The proposed solution is based on three different methods to solve the ellipse detection problem in an image. Such methods take advantage of standard image processing and mathematical algorithms, which are tailored to the target application. An experimental study demonstrates the efficacy of the proposed approach, even in the presence of low computational power and limited hardware resources.
However, to be more enriching this work some observations are shown:
1.- It is necessary to consider more references that have an impact on the introduction. Minimum above 30 references.
2.- “Finally, a suitable target configuration for the robotic arm may be finally calculated to insert the gripper fingers and perform the grasping of the reel (from inside the core)”, correct the paragraph, it is named "finally" twice.
3.- “A schematic view of 213 the selection policy is illustrated in Flowchart 9.”, in this paragraph it is necessary to put the word “figure 9”.
The article is very impressive and of great contribution, therefore, I can suggest that it can be published with minor comments.
Round 2
Reviewer 1 Report
I agree with the publication of the submitted article. Suggestions for improvement have been incorporated.